# Intraoperative Surgeon-Performed Ultrasound in Complex Partial Nephrectomy: Insights from Challenging Renal Tumors

**DOI:** 10.3390/healthcare13182325

**Published:** 2025-09-17

**Authors:** Stelian Ianiotescu, Constantin Gingu, Nicoleta Sanda, Alexandru Iordache, Alexandru Dick, Ioanel Sinescu

**Affiliations:** 1Urology Department, Carol Davila University of Medicine and Pharmacy, 020021 Bucharest, Romania; dr.ianiotescu@gmail.com; 2Center of Uronephrology and Kidney Transplantation, Fundeni Clinical Institute, 258 Fundeni Street, 022328 Bucharest, Romania; c.gingu@gmail.com (C.G.); alextrooper@gmail.com (A.D.); ioanel.sinescu@umfcd.ro (I.S.); 3Department of Nephrology, Urology, Immunology and Immunology of Transplant, Dermatology, Allergology, Faculty of Medicine, Carol Davila University of Medicine and Pharmacy, 050474 Bucharest, Romania; 4General Surgery Department, Bucharest University Emergency Hospital, Carol Davila University of Medicine and Pharmacy, 010825 Bucharest, Romania

**Keywords:** intraoperative ultrasound, nephron-sparing surgery, partial nephrectomy, renal tumors, solitary kidney, robotic surgery, open surgery, renal function preservation, RENAL nephrometry score, surgical margins

## Abstract

**Introduction:** Intraoperative ultrasound (IOUS) is increasingly utilized in nephron-sparing surgery for its ability to provide real-time, high-resolution imaging that enhances tumor localization and resection accuracy. Its role becomes particularly important in anatomically complex cases such as endophytic, multifocal, or recurrent renal tumors, as well as in patients with a solitary kidney. **Methods:** We conducted a retrospective analysis of 152 patients who underwent partial nephrectomy for localized renal tumors between January 2019 and December 2024. Patients were divided into two groups: Group A (n = 24) included patients with a solitary surgical kidney or tumor recurrence; Group B (n = 128) included patients with a contralateral functional kidney. IOUS was used in 31 cases (20%). Demographic, perioperative, and oncological outcomes were compared, with specific attention to the use and impact of IOUS. **Results:** IOUS was significantly more common in Group A (75%) than in Group B (10%) (*p* < 0.001), reflecting its preferential use in higher-complexity surgeries. The rate of positive surgical margins was low overall, with no significant difference between the IOUS and non-IOUS groups (3.2% vs. 1.7%; *p* = 0.54). IOUS was more frequently employed in cases involving medium/high RENAL nephrometry scores and multifocal tumors, contributing to improved intraoperative tumor delineation without increasing complication rates. **Conclusions:** IOUS enhances surgical precision and supports oncologic safety in both robotic and open partial nephrectomies, particularly in complex scenarios. Its use should be encouraged as a standard adjunct in conservative renal surgery, especially in patients with a solitary kidney, recurrent disease, or multifocal tumors.

## 1. Introduction

Renal tumors represent a major clinical challenge, accounting for approximately 3% of adult malignancies. With the increasing use of imaging, small renal masses are frequently diagnosed incidentally. While radical nephrectomy (RN) was historically the standard of care, the paradigm has shifted toward nephron-sparing surgery (NSS) to preserve renal function without compromising oncological and functional outcomes [1,2]. This shift is particularly critical in patients with a solitary surgical or functional kidney, where radical surgery may lead to dialysis [3,4,5].

Partial nephrectomy (PN) remains technically demanding, especially for tumors with high RENAL nephrometry scores, endophytic growth, multifocality, or recurrence after prior surgery. Surgical precision in outlining tumor margins while sparing functional parenchyma is essential to balance oncologic safety and renal preservation. Intraoperative ultrasonography (IOUS) has emerged as a valuable imaging modality in surgical practice due to its portability, real-time imaging capability, lack of ionizing radiation, and minimal need for patient preparation. Unlike conventional imaging techniques, IOUS enables the direct application of the transducer onto the organ of interest, providing superior image resolution unaffected by intervening tissues such as bone, air, or subcutaneous fat [1,3].

Currently, IOUS is widely utilized for identifying known lesions, evaluating resection boundaries, and detecting subclinical or small lesions, particularly in hepatic, renal, and pancreatic surgery. The addition of Doppler modalities, color and pulsed, enhances vascular mapping and perfusion assessment, notably in transplant surgery. Its utility has further expanded in minimally invasive platforms, including laparoscopic, endoscopic, and robotic-assisted procedures [1,4,6]. IOUS has emerged as a critical tool in modern nephron-sparing surgery, especially in the management of small renal masses with complex anatomical features. Its use allows for accurate localization of endophytic tumors (Figure 1 and Figure 2), enabling maximal preservation of healthy renal parenchyma during resection [7,8,9,10,11]. Furthermore, intraoperative ultrasound plays an essential role in cases of irregular or multifocal tumors (Figure 4) by providing precise definition of tumor margins [4,5]. Emerging technologies such as contrast-enhanced ultrasound (CEUS) and ultrasound elastography are enhancing the diagnostic capabilities of IOUS, offering real-time functional and mechanical tissue characterization [12].

At present, the upper limit of indications for partial nephrectomy (PN) remains undefined and is largely determined by the surgeon’s expertise and clinical judgment. The variability in selecting PN versus radical nephrectomy (RN) increases with tumor size. Surgeons with a strong commitment to nephron-sparing surgery tend to broaden the indications for PN, whereas others, concerned about higher perioperative morbidity or skeptical of the clinical significance of a moderate decline in renal function, are more inclined to perform RN regardless of tumor size. We hypothesized that, in patients with tumors staged cT2 or higher, the benefits of preserving renal function outweigh the risks of perioperative complications. If confirmed, this approach could support the extension of PN indications to larger renal tumors, with the potential to improve postoperative renal outcomes in this patient population. Moreover, the use of intraoperative ultrasonography (IOUS) may provide additional value in such complex and technically demanding cases, by enhancing tumor localization, facilitating accurate resection margins, and supporting safe nephron-sparing surgery. Based on our experience, IOUS allows precise delineation of tumor borders, thereby minimizing the unnecessary excision of healthy renal parenchyma and ensuring that the remaining parenchyma is sufficient to maintain stable renal function and avoid progression to dialysis. Furthermore, in imperative cases, such as tumors in a solitary surgical or functional kidney, PN becomes not only the preferred but also the mandatory approach, where IOUS can be decisive for surgical success.

Existing studies suggest that intraoperative ultrasound (IOUS) may contribute to reducing positive surgical margin (PSM) rates and improving renal functional preservation, but the available evidence remains limited, heterogeneous, and often based on small or single-center series [9,10,11]. Several authors have underlined that, despite advances in preoperative imaging such as CT and MRI, intraoperative delineation of tumor margins and identification of satellite lesions remain significant challenges, particularly in endophytic, hilar, multifocal, or recurrent tumors. Recent literature therefore emphasizes the importance of integrating refined intraoperative imaging tools into nephron-sparing surgery, especially for high-risk renal masses, where surgical precision and parenchymal preservation are equally critical [13]. Intraoperative imaging should not only support surgical navigation but also actively guide intraoperative decision-making, improving both oncologic safety and postoperative functional outcomes. In this context, our study was designed to assess the clinical utility of IOUS in complex partial nephrectomies performed via both robotic-assisted and open approaches. The primary objective was to determine whether IOUS use is associated with reduced PSM rates, while the secondary objectives were to explore its impact on perioperative parameters (warm ischemia time, estimated blood loss, complication rates) and on postoperative functional outcomes measured by changes in estimated glomerular filtration rate (eGFR). We hypothesized that IOUS provides a measurable intraoperative advantage by enhancing tumor localization, improving the accuracy of resection planes, and maximizing renal parenchymal preservation, particularly in demanding scenarios such as solitary surgical kidney, solitary functional kidney, tumor recurrence, or multifocal disease.

## 2. Materials and Methods

After institutional review board approval: 6 May 2025, we retrospectively analyzed 152 patients who underwent partial nephrectomy for localized renal tumors between January 2019 and December 2024. Among them, 110 patients underwent robot-assisted laparoscopic partial nephrectomy, while 42 patients were treated with the open surgical approach. Patients were stratified into two groups: Group A included those with a solitary surgical kidney or tumor recurrence in a solitary kidney, and Group B included patients with a functional contralateral kidney.

Patients were included in the intraoperative ultrasound (IOUS) group when preoperative CT or MRI demonstrated endophytic tumors, multifocal disease, or local recurrence, all considered high-complexity features; the final decision to employ IOUS was made by the surgeon.

Given the relatively small sample size, multivariable logistic regression was not feasible. Instead, univariate analyses were performed to assess associations between clinical and oncological variables and IOUS use. Chi-square or Fisher’s exact test was applied for categorical variables, and Student’s t-test or Mann–Whitney U test was used for continuous variables, as appropriate. Variables examined included patient age, sex, surgical approach (open vs. robotic), clamping time, pathological T stage, presence of multiple tumors, and IOUS use. All tests were two-sided, and a *p*-value < 0.05 was considered statistically significant.

In robot-assisted laparoscopic surgeries, the I12C4f (9066) transducer (Figure 3, Figure 4 and Figure 5) was used, while in open surgeries, a linear soft-tissue probe compatible with the BK Medical 5000 system (BK Medical, Boston, MA, USA) was employed (Figure 6). Data were collected retrospectively and analyzed to compare image clarity, accuracy of tumor margin delineation, and intraoperative outcomes.

While different probes were used depending on the surgical approach, both provided high-resolution imaging of tumor margins. Based on our analysis, probe type did not appear to influence the oncological and functional outcomes.

## 3. Results

Baseline demographic and functional characteristics are summarized in Table 1. Patients in Group A (solitary surgical kidney or recurrent tumor) were older on average (64 vs. 58 years, *p* = 0.028) and had significantly higher baseline creatinine and lower eGFR compared to Group B (*p* < 0.001). At 6 months, renal function remained significantly worse in Group A. Regarding tumor complexity, medium and high RENAL scores were more frequent in Group A (46% vs. 11%, *p* < 0.001).

Perioperative outcomes are detailed in Table 2. Open surgery was more common in Group A (54% vs. 23%, *p* = 0.002), while robotic surgery predominated in Group B. Mean blood loss was significantly higher in Group A (203 vs. 125 mL, *p* = 0.011), with lower postoperative hemoglobin (11.2 vs. 12.1 g/dL, *p* = 0.042) and a higher transfusion rate (17% vs. 3%, *p* = 0.022). The length of hospital stay was longer in Group A (7.3 vs. 5.3 days, *p* = 0.024). Complications were also more frequent in Group A (25% vs. 5%, *p* = 0.004). Importantly, intraoperative ultrasound (IOUS) was used in 31 of 152 cases (20%), with a significantly higher frequency in Group A (75% vs. 10%, *p* < 0.001).

Oncological results are shown in Table 3. Pathological staging differed between the groups, with more advanced tumors (T2–T3) in Group A (25% vs. 4.7%, *p* = 0.004). Multiple tumors were also more common in Group A (25% vs. 2%, *p* < 0.001). Positive surgical margins were rare overall (2%), with no significant difference between groups (*p* = 0.41).

Intraoperative ultrasound was used in 75% of Group A cases compared with only 10% of Group B, representing a 7.5-fold higher utilization rate. This preferential use reflects the greater anatomical complexity of solitary kidney and recurrent tumor cases, where IOUS proved beneficial in guiding tumor localization and resection. Positive surgical margins were identified in only 3 patients across the entire cohort (1 in the IOUS group [3.2%] and 2 in the non-IOUS group [1.7%], *p* = 0.54). Although IOUS was more frequently used in complex cases with higher RENAL scores, multifocality, or solitary kidney, its use did not translate into a higher rate of PSMs, suggesting that IOUS facilitated safe resections even in anatomically demanding scenarios.

The RENAL nephrometry score distribution further highlighted that IOUS was preferentially used in more complex tumors: 39% of IOUS cases involved medium or high complexity lesions (RENAL ≥ 7), compared with only 11% in the non-IOUS group (*p* < 0.001). Moreover, the IOUS group presented with a significantly higher rate of multifocal tumors (19% vs. 2.5%, *p* = 0.003), a trend toward larger tumor size (4.4 vs. 3.9 cm, *p* = 0.14), and more frequent use of the open approach (61% vs. 19%, *p* < 0.001).

Warm ischemia time was comparable between IOUS and non-IOUS cases (22.9 ± 4.3 vs. 23.1 ± 3.9 min, *p* = 0.22). Immediate postoperative complications occurred at similar rates between open (33%) and robotic (25%) approaches (*p* = 0.99). Specific complications included trocar-site hematomas (n = 2, robotic group), wound infection (n = 1, open group), parietal hematoma requiring reintervention (n = 1, open group), and one case of postoperative anuria requiring temporary dialysis (n = 1, open group). In the latter case, creatinine rose to 4.55 mg/dL; after hemodialysis and nephrological management, diuresis progressively recovered, and creatinine improved to 1.29 mg/dL at discharge. No patient required chronic dialysis.

Renal function followed the expected postoperative course: serum creatinine increased and eGFR decreased immediately after surgery, but both parameters improved during follow-up. Complications were classified according to Clavien–Dindo. In the robotic group, two grade II complications occurred. In the open group, two grade IIIb complications (surgical reintervention for bleeding and ureteral stent placement for fistula) and one grade IVa complication (dialysis for postoperative anuria) were recorded.

Intraoperative ultrasound (IOUS) was used in 31 of 152 cases (20%). Its use varied significantly between the two study groups, being more frequent in Group A (patients with a solitary surgical kidney or recurrent tumor in a solitary kidney), where it was utilized in 18 of 24 cases (75%), compared to 13 of 128 cases (10%) in Group B (*p* < 0.001). This reflects a preferential use of IOUS in anatomically or functionally complex scenarios.

Intraoperative ultrasound was used in 75% of Group A cases compared with only 10% of Group B, representing a 7.5-fold higher utilization rate. This preferential use reflects the greater anatomical complexity of solitary kidney and recurrent tumor cases, where IOUS proved beneficial in guiding tumor localization and resection.

Positive surgical margins (PSMs) were identified in only 3 patients across the entire cohort (1 in the IOUS group [1/31, 3.2%] and 2 in the non-IOUS group [2/121, 1.7%], *p* = 0.54). Although IOUS was more frequently used in complex cases with higher RENAL scores, multifocality, or solitary kidney, its use did not translate into a higher rate of PSMs, suggesting that IOUS facilitated safe resections even in anatomically demanding scenarios.

The RENAL nephrometry score distribution revealed a significantly higher proportion of medium and high-complexity tumors in Group A compared with Group B. These findings highlight that patients in Group A faced a substantially greater surgical challenge due to the predominance of more anatomically complex renal masses.

Importantly, IOUS was predominantly used in cases with higher surgical complexity, as reflected by the RENAL nephrometry score distribution:

In the IOUS group, 12 of 31 tumors (39%) were of medium or high complexity (RENAL score ≥ 7).

In contrast, only 13 of 121 tumors (11%) in the non-IOUS group were classified as medium/high complexity (*p* < 0.001).

Additionally, the IOUS group had the following characteristics:

A significantly higher rate of multifocal tumors: 6/31 (19%) vs. 3/121 (2.5%), *p* = 0.003.

A higher mean tumor size (4.4 vs. 3.9 cm), though this was not statistically significant (*p* = 0.14).

More frequent use of the open surgical approach (19/31 cases, 61%) compared to the non-IOUS group (23/121 cases, 19%), *p* < 0.001.

Warm ischemia time was comparable between the IOUS and non-IOUS groups, with no statistically significant difference observed (22.9 ± 4.3 vs. 23.1 ± 3.9 min, *p* = 0.22).

Immediate postoperative complications occurred at similar rates in the open (33%) and robotic (25%) approaches (*p* = 0.99). Specific complications included trocar-site hematomas (n = 2, robotic group), wound infection (n = 1, open group), parietal hematoma requiring reintervention (n = 1, open group), and one case of postoperative anuria requiring temporary dialysis (n = 1, open group). In the latter case, creatinine rose to 4.55 mg/dL; after hemodialysis and nephrological management, diuresis progressively recovered (400 → 800 → 1200 → 2000 mL/day), and creatinine improved to 1.29 mg/dL at discharge. No patient required chronic dialysis.

Renal function followed the expected postoperative course: serum creatinine increased and eGFR decreased immediately after surgery but both parameters improved during follow-up. Complications were classified according to Clavien–Dindo. In the robotic group, two grade II complications occurred. In the open group, two grade IIIb complications (surgical reintervention for bleeding and ureteral stent placement for fistula) and one grade IVa complication (dialysis for postoperative anuria) were recorded.

## 4. Discussion

Intraoperative ultrasound represents one of the most impactful technical innovations in the field of nephron-sparing surgery. It has revolutionized conservative renal tumor management by providing a level of intraoperative precision and safety previously unattainable with traditional open or standard laparoscopic techniques. Understanding the value of this technology is essential to appreciate its clinical impact, particularly in highly sensitive scenarios such as surgeries involving a solitary kidney or tumor recurrence.

One of the main advantages of intraoperative ultrasound is its ability to provide three-dimensional mapping of the tumor in relation to healthy parenchyma, the collecting system, and vascular structures [4]. This enables the surgeon to define the resection plane with high accuracy, avoid critical structures, and minimize excision of functional tissue—an essential consideration in nephron-sparing procedures, where renal function preservation is as important as oncologic control.

In completely endophytic tumors—those located entirely within the renal parenchyma and not visible externally—intraoperative ultrasound becomes indispensable. In the absence of real-time imaging, localization of such tumors is extremely difficult, increasing the risk of incorrect incision, incomplete resection, or inadvertent damage to adjacent healthy tissue. Ultrasound guidance allows precise identification of tumor depth and its relationship to the collecting system and segmental vessels, even when the kidney appears macroscopically normal.

In cases of complex tumors, such as those with high RENAL nephrometry scores, intraoperative ultrasound enhances real-time surgical planning. It assists in choosing safe and efficient resection planes, minimizing the risk of positive surgical margins or unnecessary parenchymal loss [5].

Another key benefit is its ability to detect multifocal disease. While preoperative imaging techniques such as CT or MRI may detect multiple lesions, some small satellite nodules may be missed. Intraoperative ultrasound permits systematic scanning of the renal parenchyma during surgery and enables identification of additional lesions, contributing to oncologic completeness of the intervention.

In robotic-assisted partial nephrectomy, intraoperative ultrasound integrates seamlessly into the surgical workflow. Robotic instruments can manipulate compatible ultrasound probes without significantly interrupting the procedure. This allows the surgeon to alternate between standard endoscopic views and ultrasound imaging directly from the console, enhancing operative focus and efficiency.

Modern surgical platforms also enable augmented reality features by overlaying ultrasound images on the operative field, facilitating orientation and accuracy [6].

Although IOUS was preferentially applied in patients with higher surgical complexity, including solitary kidneys, recurrences, multifocal tumors, and medium-to-high RENAL scores, perioperative and functional outcomes were not inferior compared with the non-IOUS group. Despite the greater baseline risk profile, postoperative eGFR decline and complication rates were comparable, and no patient required chronic dialysis. This observation suggests that IOUS may play a compensatory role, allowing surgeons to preserve functional parenchyma and avoid morbidity even in technically demanding cases. While causality cannot be established due to the retrospective design and limited sample size, these findings support the hypothesis that IOUS facilitates safe nephron-sparing surgery in anatomically and functionally challenging situations.

Ultrasound also aids in assessing tumor vascularization, informing decisions about ischemia management. When a lesion is supplied by an isolated arterial branch, segmental clamping may be performed to preserve perfusion to the remaining kidney. Although angiography with contrast injection in interventional radiology can also provide vascular mapping, IOUS offers the advantage of real-time intraoperative guidance, without the need for contrast administration or additional radiation exposure, which makes it particularly valuable in the surgical setting. A further application is verifying complete tumor resection. Post-excision ultrasound allows for assessment of the resection bed to ensure no residual tumor or satellite nodules remain, which can improve oncologic outcomes and reduce the need for re-intervention [12,14,15].

In cases of tumor recurrence, intraoperative ultrasound becomes even more valuable. Postoperative scarring and fibrosis distort normal anatomy, making visual and tactile identification unreliable. Real-time ultrasound enables differentiation between scar tissue and tumor, helping to safely guide resection in anatomically distorted fields.

From a functional standpoint, intraoperative ultrasound has been associated with lower postoperative renal insufficiency rates and better preservation of glomerular filtration rate at 6–12 months, underscoring its importance in minimizing healthy tissue loss [7,16].

There are also notable educational advantages: it helps training surgeons better understand intrarenal anatomy, develop dissection precision, and appreciate variability in renal vasculature and collecting system configurations.

Overall, intraoperative ultrasound has transformed the paradigm of partial nephrectomy. It allows a shift from intuition-based dissection to precision-guided surgery, reducing oncologic and functional risks while optimizing each step of the procedure. Our findings suggest that IOUS represents a promising adjunct in complex conservative renal surgery and may, in the future, be integrated more systematically into standard practice pending confirmation in prospective studies [17,18].

This study has several limitations. Its retrospective design and the selective use of intraoperative ultrasound in complex cases may introduce selection bias. Moreover, the small number of positive surgical margins limits the ability to draw firm conclusions regarding the oncological impact of IOUS. Future prospective studies with larger cohorts are needed to clarify the independent role of IOUS, particularly in high-complexity renal tumors.

IOUS was not applied systematically but rather based on intraoperative complexity, surgeon’s judgment, and specific tumor characteristics (endophytic location, multifocality, hilar involvement, or recurrent tumors). This selective use may have introduced bias, as reflected by its more frequent use in Group A (solitary kidney/recurrence) compared with Group B. This aspect is addressed in the Discussion.

## 5. Conclusions

Intraoperative ultrasound is a promising adjunct in complex partial nephrectomy. Although our data are preliminary and limited by retrospective design and small sample size, IOUS appears to enhance surgical precision and renal preservation in high-risk cases such as solitary kidneys and recurrent tumors. Future prospective, multicenter studies are required to confirm its independent value and determine whether IOUS should be systematically adopted as an adjunct tool in complex nephron-sparing surgery.

## Figures and Tables

**Figure 1 healthcare-13-02325-f001:**
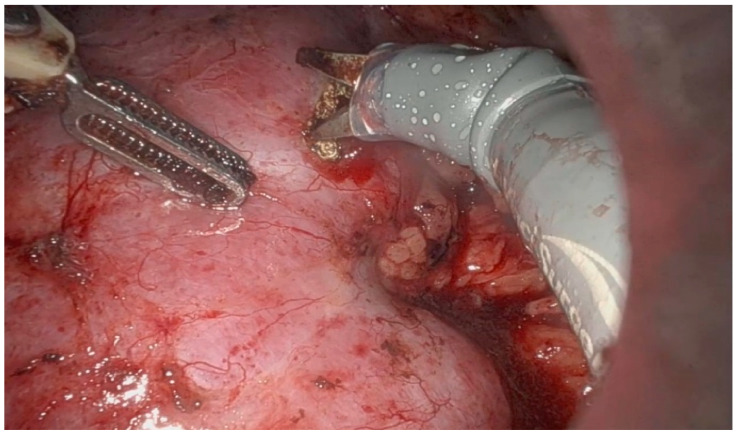
Endophytic renal tumor (outlining of tumor margins).

**Figure 2 healthcare-13-02325-f002:**
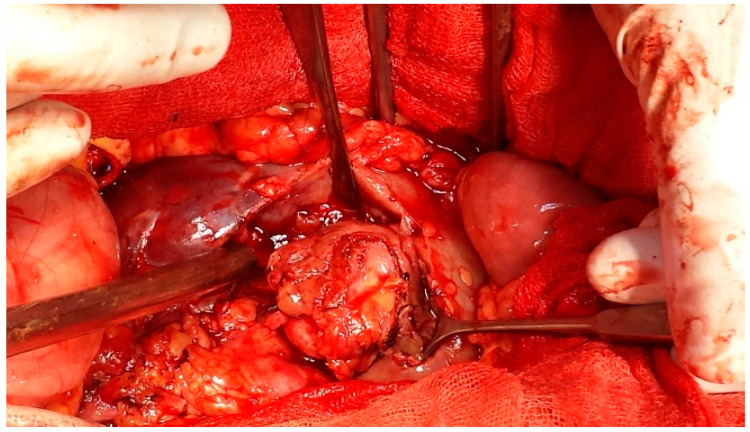
Renal sinus tumor—open surgical approach.

**Figure 3 healthcare-13-02325-f003:**
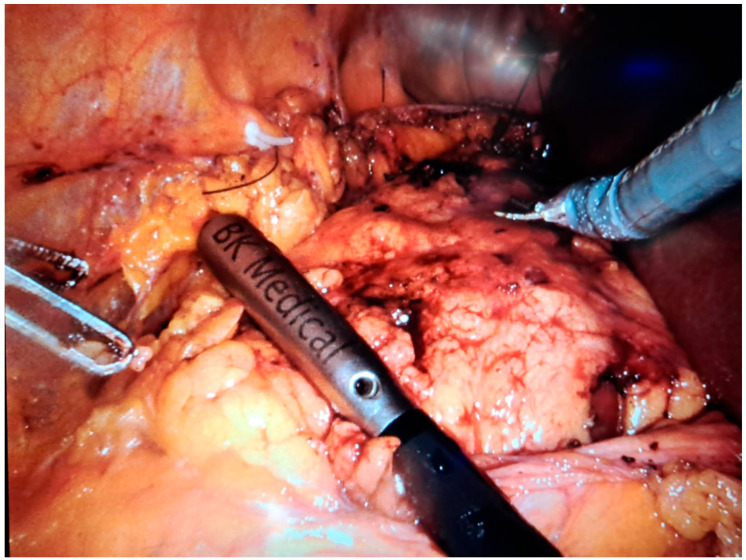
Intraoperative ultrasound imaging performed during a robot-assisted laparoscopic partial nephrectomy—identification of an endophytic tumor hidden beneath adherent perinephric toxic fat.

**Figure 4 healthcare-13-02325-f004:**
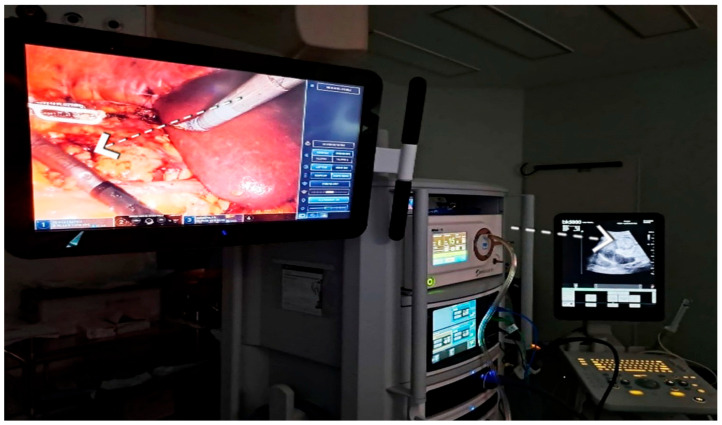
Intraoperative ultrasound during robot-assisted laparoscopic approach—endophytic tumor identifiable exclusively on the ultrasound monitor.

**Figure 5 healthcare-13-02325-f005:**
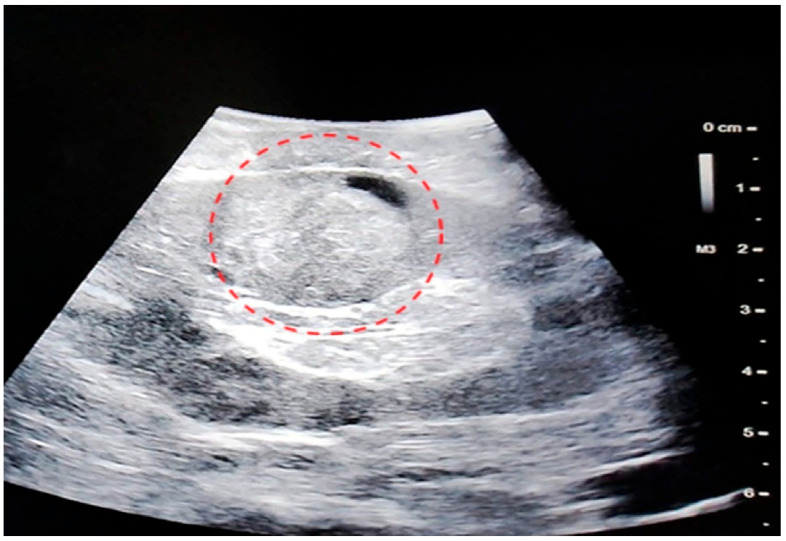
Ultrasound appearance of an endophytic lesion. Red circle highlighting the ultrasound appearance of an endophytic renal lesion.

**Figure 6 healthcare-13-02325-f006:**
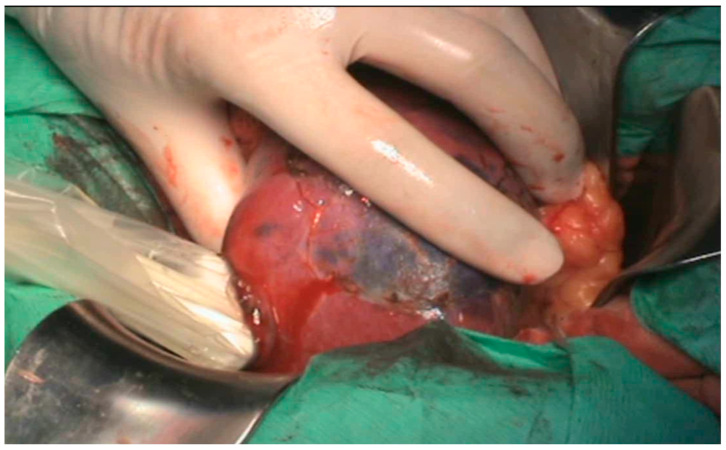
Intraoperative ultrasound—open surgical approach. (no tumor expression).

**Table 1 healthcare-13-02325-t001:** Baseline characteristics of patients undergoing partial nephrectomy.

Variable	Global (N = 152)	Group A (N = 24)	Group B (N = 128)	*p*-Value
Age, Mean (SD), years	59 (12)	64 (12)	58 (12)	0.028
Sex—Female (%)	56 (37%)	8 (33%)	48 (38%)	0.70
Sex—Male (%)	96 (63%)	16 (67%)	80 (63%)	0.70
Baseline Creatinine, Mean (SD), mg/dL	1.07 (0.56)	1.76 (0.78)	0.94 (0.40)	<0.001
6-month Follow-up Creatinine, Mean (SD), mg/dL	0.98 (0.36)	1.46 (0.45)	0.89 (0.25)	<0.001
Baseline eGFR, Mean (SD), mL/min/1.73 m^2^	78 (24)	46 (25)	83 (19)	<0.001
6-month Follow-up eGFR, Mean (SD), mL/min/1.73 m^2^	80 (23)	52 (25)	85 (19)	<0.001
RENAL—Low (%)	127 (83%)	13 (54%)	114 (89%)	<0.001
RENAL—Medium (%)	21 (14%)	7 (29%)	14 (11%)	<0.001
RENAL—High (%)	4 (3%)	4 (17%)	0 (0%)	<0.001

**Table 2 healthcare-13-02325-t002:** Perioperative outcomes of patients undergoing partial nephrectomy.

Variable	Global (N = 152)	Group A (N = 24)	Group B (N = 128)	*p*-Value
Surgical Approach—Open (%)	42 (28%)	13 (54%)	29 (23%)	0.002
Surgical Approach—Robotic (%)	110 (72%)	11 (46%)	99 (77%)	0.002
Clamping Time, Mean (SD), min	22.9 (4.3)	21.6 (5.8)	23.1 (3.9)	0.22
Blood Loss, Mean (SD), mL	138 (114)	203 (133)	125 (106)	0.011
Postoperative Hemoglobin, Mean (SD), g/dL	11.93 (1.68)	11.23 (1.79)	12.07 (1.63)	0.042
Transfusions Yes, n (%)	8 (5%)	4 (17%)	4 (3%)	0.022
Complications Present (%)	12 (8%)	6 (25%)	6 (5%)	0.004
Tumor Size, Mean (SD), cm	4.01 (1.55)	4.55 (2.36)	3.91 (1.34)	0.21
Length of Stay, Mean (SD), days	5.64 (2.34)	7.33 (4.04)	5.32 (1.70)	0.024
Intraoperative Ultrasound Use, n (%)	31 (20%)	18 (75%)	13 (10%)	<0.001

**Table 3 healthcare-13-02325-t003:** Oncological outcomes of patients undergoing partial nephrectomy.

Variable	Global (N = 152)	Group A (N = 24)	Group B (N = 128)	*p*-Value
Pathological T Stage, n (%)				0.004
T1	140 (92%)	18 (75%)	122 (95%)	
T2–T3	12 (7.9%)	6 (25%)	6 (4.7%)	
Multiple Tumors Yes, n (%)	9 (6%)	6 (25%)	3 (2%)	<0.001
Positive Margins Yes, n (%)	3 (2%)	1 (4%)	2 (2%)	0.41

## Data Availability

The original contributions presented in this study are included in the article. Further inquiries can be directed to the corresponding authors.

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
