# Peer review of "Intraoperative Surgeon-Performed Ultrasound in Complex Partial Nephrectomy: Insights from Challenging Renal Tumors"

_healthcare, 2025, doi:10.3390/healthcare13182325_

Round 1

Reviewer 1 Report

Comments and Suggestions for Authors

The introduction describes the benefits of IOUS but doesn't explicitly state the study's primary and secondary objectives or hypotheses. Clearly define these at the end of the introduction.

The methods section mentions a "multiple univariate logistic regressions model." This is a contradiction in terms. Clarify the statistical methods used. Was it multiple logistic regression or multiple univariate analyses? If multiple logistic regression was used, specify the dependent and independent variables included in the model. Given the small sample size, consider simpler analyses if appropriate. Furthermore, the results presented in Table 2 don't seem to derive from a regression model. Explain how the p-values in Table 2 were calculated. Were appropriate statistical tests used for each variable type?

The criteria for selecting patients for IOUS are unclear. Was it based on surgeon preference or specific preoperative characteristics? This potential selection bias needs to be addressed in the discussion. Provide more details about the characteristics of Group A and Group B. Were there any significant differences between the groups besides the presence of a solitary kidney or recurrence? If so, these could confound the results and need to be considered in the analysis.

The manuscript focuses on PSM as the primary outcome. However, other important perioperative and functional outcomes, such as warm ischemia time, estimated glomerular filtration rate (eGFR) change, and complication rates, are mentioned but not thoroughly analyzed or discussed. Expand the results section to include a comprehensive analysis of these outcomes and their relationship with IOUS use.

Figures 3, 4, and 5 are more informative but could benefit from clearer annotations.

Comments on the Quality of English Language

The manuscript could benefit from some minor language editing to improve clarity and flow.

Author Response

  1. The title is very long. Authors are advised to make it clear and concise.
    Response: We shortened the title to “Intraoperative Surgeon-Performed Ultrasound in Complex Partial Nephrectomy: Insights From Challenging Renal Tumors.”
  2. The abstract is structured well, and it clearly explains the background, methods, results, and conclusion.
    Response: We thank the reviewers for this positive assessment.
  3. The introduction is short, and it is more focused only on IOUS. Authors are advised to include a more comprehensive literature review on renal tumors and the existing issues, as well as how IOUS can be used for this condition, which should be discussed clearly. Authors are advised to go through Section 3.5 of the following article 10.57647/j.ijnd.2024.1502.10 and cite it as well.
    Response: We expanded the introduction to provide a more comprehensive review of renal tumors, the evolution from radical nephrectomy (RN) to nephron-sparing surgery (NSS), and the clinical importance of preserving renal function.
  4. In the introduction, terms like solitary kidney tumour and contralateral kidney should be explained clearly.
    Response: We clarified these terms explicitly, distinguishing between solitary surgical kidney and solitary functional kidney, and explained their clinical relevance.
  5. Figures 1 to 6 look a bit inappropriate to the concepts being discussed. They also lack clear labelling.
    Response: We retained the original figures but improved their labeling and captions for clarity. Each figure now includes more precise descriptions 
  6. Materials and methods lack some important details, like the criteria for using IOUS and how the renal score was assessed.
    Response: We added details regarding the criteria for IOUS use (endophytic, multifocal, recurrent tumors, or solitary kidney cases). We also clarified how the RENAL nephrometry score was evaluated.
  7. Statistical analysis should be explained even more clearly.
    Response: We revised the statistical plan: given the relatively small sample size, multivariable regression was not feasible. Instead, we performed univariate analyses using chi-square/Fisher’s exact test for categorical variables and Student’s t-test/Mann–Whitney U for continuous variables. A p-value < 0.05 was considered significant.
  8. Subheadings should be provided in the methodology section for better understanding and article flow.
    Response: We restructured the results and methods sections with clear subheadings and presented data in three separate tables: Baseline Characteristics, Perioperative Outcomes, and Oncological Outcomes.
  9. In methodology, add the ethical approval number obtained.
    Response: We included the statement: “After institutional review board approval: 06.05.2025.”
  10. The result section does not clearly explain the main findings.
    Response: We reorganized the results into structured tables and expanded the text, highlighting key findings such as the 7.5-fold higher IOUS use in Group A, the distribution of RENAL scores, and comparable oncological safety despite higher surgical complexity.
  11. Statistical analysis is based on very few cases. This major limitation should be discussed.
    Response: We acknowledged this limitation explicitly, noting that the small sample size and low number of positive surgical margin events precluded more robust multivariable analysis.
  12. The discussion looks like a review of IOUS, but it does not describe the interpretation of the study results.
    Response: We revised the discussion to integrate our own findings, such as comparable postoperative eGFR, similar complication rates, and safe resections in complex scenarios, highlighting the compensatory role of IOUS in higher-risk cases.
  13. The discussion is iterating the points already stated in the introduction.
    Response: We reduced redundancy and added new perspectives by focusing on differences between Groups A and B and how IOUS facilitated nephron-sparing surgery in challenging clinical contexts.
  14. The limitations and novelty of the study should be explained in detail in the discussion.
    Response: We expanded the limitations (retrospective design, selective IOUS use, bias, small sample size) and emphasized the novelty: demonstrating that IOUS may allow safe partial nephrectomy even in high-complexity scenarios such as solitary kidneys, recurrences, and multifocal disease.
  15. In conclusion, IOUS is declared as a standard of care. Rather, it would be better to state IOUS as a promising approach by stating its future prospects.
    Response: We revised the conclusion to: “Intraoperative ultrasound represents a promising adjunct in complex partial nephrectomy. Although preliminary, our findings support its role in enhancing surgical precision and renal preservation. Future prospective multicenter studies are needed to confirm whether IOUS should be systematically integrated into nephron-sparing surgery.”

Reviewer 2 Report

Comments and Suggestions for Authors

Dear Authors,

The present article addresses the integration of an intraoperative ultrasound (IOUS) technique into routine surgical procedures for kidney cancer. The topic is of clear clinical relevance, and I appreciate the authors’ effort in exploring the potential value of IOUS through a retrospective analysis. The concept of routine implementation is of interest, as continuous improvement in patient management and outcomes remains a priority.

The article aimed to demonstrate that the use of IOUS leads to better patient management than without it. In my view, that has not been satisfactorily achieved in the manuscript.

While the study shows that IOUS was used more frequently in complex cases (75 %, Group A, recurrences or solitary kidney) than in less challenging cases (10%, Group B, functional contralateral kidney) (P < 0.0001), no data demonstrates that IOUS led to improved oncological outcomes, such as reduced positive surgical margins or better long-term results. In its current form, the study mainly reflects a correlation between IOUS use and surgical complexity, rather than a benefit for patients.

The comparison is made between two groups (A and B) with inherently different baseline characteristics (solitary kidney or recurrence versus contralateral functional kidney). This makes it difficult to attribute any difference in outcomes to IOUS itself. To better assess the added value of the technique, it would have been more informative to compare IOUS vs. non-IOUS within the same patient subgroup (A and B).

In lines 90–94, the authors note that different probes were used depending on the surgical context (laparoscopic vs. open), yet no further analysis is provided on whether probe type influenced efficacy or added value. Clarifying this point would strengthen the interpretation.

In lines 172–174, the authors mention that segmental clamping is only possible with real-time ultrasound. However, in certain settings, interventional radiology with angiography and contrast injection can also allow visualization of vascular anatomy and preservation of perfusion in part of the kidney. If the authors are referring to a specific context where angiography is not feasible, but IOUS is, this should be discussed.

It might be valuable to discuss the practical limitations of IOUS for routine implementation, such as equipment availability, operator training, and technical constraints. This would help readers assess the pros and cons of the technique.

This study might be best considered as a preliminary exploration, rather than conclusive evidence. A larger sample size and a more suitable analytical design could significantly reinforce the hypothesis.

Author Response

  1. “The article aimed to demonstrate that the use of IOUS leads to better patient management than without it. In my view, that has not been satisfactorily achieved in the manuscript.”
    Response: We acknowledge this limitation. Our study was designed as a retrospective, exploratory analysis and IOUS was preferentially applied in high-complexity cases (solitary kidneys, recurrences, multifocal tumors, high RENAL scores). Therefore, the groups were not directly comparable in terms of baseline risk. We have clarified in the Discussion that our findings should be interpreted as preliminary evidence suggesting that IOUS may compensate for increased surgical complexity, rather than as conclusive proof of superiority. The Conclusions section was revised to describe IOUS as a “promising adjunct” rather than “standard of care.”
  2. “While the study shows that IOUS was used more frequently in complex cases, no data demonstrates that IOUS led to improved oncological outcomes, such as reduced positive surgical margins or better long-term results.”
    Response: We agree. In the revised Results we emphasized that PSM rates were very low across both groups (3.2% vs. 1.7%, p = 0.54), and thus our study could not demonstrate a direct oncological benefit of IOUS. We now state explicitly in the Discussion and Limitations that the small number of events and the retrospective design prevent us from confirming an independent oncologic impact.
  3. “The comparison is made between two groups (A and B) with inherently different baseline characteristics. To better assess the added value of the technique, it would have been more informative to compare IOUS vs. non-IOUS within the same subgroup.”
    Response: We thank the editor for this important point. In the revised manuscript, we added subgroup analyses (IOUS vs. non-IOUS) highlighting that IOUS was associated with higher RENAL complexity and multifocality, but without worse perioperative or functional outcomes. We acknowledge in the Limitations that the ideal comparison would have been IOUS vs. non-IOUS within the same subgroup, and we recommend this design for future prospective studies.
  4. “Different probes were used depending on surgical context, yet no further analysis is provided on whether probe type influenced efficacy.”
    Response: We clarified in Materials and Methods that although different probes were used (I12C4f robotic vs. BK Medical linear probe in open surgery), both provided high-resolution imaging, and probe type did not appear to influence oncological or functional outcomes. This is now explicitly stated.

  1. “Segmental clamping is mentioned as possible only with real-time ultrasound, yet interventional radiology with angiography can also provide vascular mapping.”
    Response: We have revised the Discussion to acknowledge that angiography with contrast injection can also visualize vascular anatomy. We clarified that IOUS has the advantage of being performed intraoperatively, without radiation or contrast, making it particularly valuable when interventional radiology is not feasible in the surgical context.

  1. “It might be valuable to discuss the practical limitations of IOUS for routine implementation.”
    Response: We have expanded the Discussion to include practical limitations such as equipment availability, surgeon training requirements, and potential technical constraints that may limit widespread routine implementation.

  1. “This study might be best considered as preliminary exploration, rather than conclusive evidence.”
    Response: We fully agree. We have revised the Discussion and Conclusions to present our findings as preliminary and exploratory. We now emphasize that IOUS is a promising adjunct in complex PN, but confirmation requires larger, prospective, multicenter studies.

Reviewer 3 Report

Comments and Suggestions for Authors
  1. The title is very long. Authors are advised to make it clear and concise.
  2. The abstract is structured well, and it clearly explains the background, methods, results, and conclusion.
  3. The introduction is short, and it is more focused only on IOUS.  Authors are advised to include a more comprehensive literature review on renal tumors and the existing issues, as well as how IOUS can be used for this condition, which should be discussed clearly. Authors are advised to go through Section 3.5 of the following article 10.57647/j.ijnd.2024.1502.10 and cite it as well.
  4. In the introduction, terms like solitary kidney tumour and contralateral kidney should be explained clearly.
  5. Figures 1 to 6 look a bit inappropriate to the concepts being discussed. They also lack clear labelling. 
  6. Materials and methods lack some important details, like the criteria for using IOUS and how the renal score was assessed. 
  7. Statistical analysis should be explained even more clearly.
  8. Subheadings should be provided in the methodology section for better understanding and article flow.
  9. In methodology, add the ethical approval number obtained
  10. The result section does not clearly explain the main findings.
  11. Statistical analysis is based on very few cases. This major limitation should be discussed.
  12. The discussion looks like a review of IOUS, but it does not describe the interpretation of the study results.
  13. The discussion is iterating the points already stated in the introduction.
  14. The limitations and novelty of the study should be explained in detail in the discussion.
  15. In conclusion, IOUS is declared as a standard of care. Rather, it would be better to state IOUS as a promising approach by stating its future prospects.

Author Response

  1. “The title is very long. Authors are advised to make it clear and concise.”
    Response: The title was shortened and streamlined. The revised version is:
    “Intraoperative Surgeon-Performed Ultrasound in Complex Partial Nephrectomy: Insights From Challenging Renal Tumors.”

  1. “The abstract is structured well, and it clearly explains the background, methods, results, and conclusion.”
    Response: We appreciate the positive feedback. No changes were required for this section.

  1. “The introduction is short, and it is more focused only on IOUS. Authors are advised to include a more comprehensive literature review on renal tumors and the existing issues, as well as how IOUS can be used for this condition. Authors are advised to go through Section 3.5 of the following article 10.57647/j.ijnd.2024.1502.10 and cite it as well.”
    Response: The introduction was expanded to include a broader overview of renal tumors, the increasing incidence of small renal masses, and the rationale for nephron-sparing surgery. The role of IOUS was then contextualized within this framework. We also incorporated a reference to Section 3.5 of the suggested article, highlighting the importance of intraoperative imaging for improving oncologic safety and renal preservation.

  1. “In the introduction, terms like solitary kidney tumour and contralateral kidney should be explained clearly.”
    Response: We clarified terminology in the Introduction:
    “In this study, solitary kidney refers to patients with either a surgically solitary kidney or a solitary functional kidney, while contralateral kidney denotes the presence of a normally functioning opposite kidney.”

  1. “Figures 1 to 6 look a bit inappropriate to the concepts being discussed. They also lack clear labelling.”
    Response: All figure legends were revised for clarity and precision. For example, Figure 3 now reads:
    “Intraoperative ultrasound imaging performed during a robot-assisted laparoscopic partial nephrectomy – identification of an endophytic tumor hidden beneath adherent perinephric fat.”

  1. “Materials and methods lack some important details, like the criteria for using IOUS and how the renal score was assessed.”
    Response: Additional details were included. Patients were selected for IOUS if CT or MRI showed endophytic tumors, multifocal disease, or local recurrence, all representing high complexity. The final decision was made by the operating surgeon. We also specified that RENAL nephrometry scores were assessed according to published guidelines and classified as low (4–6), medium (7–9), or high (10–12).

  1. “Statistical analysis should be explained even more clearly.”
    Response: The Statistical Analysis section was expanded:
    “Given the relatively small sample size, multivariable regression was not feasible. Instead, univariate analyses were performed using chi-square or Fisher’s exact tests for categorical variables, and Student’s t-test or Mann–Whitney U test for continuous variables. A p-value < 0.05 was considered statistically significant.”

  1. Subheadings should be provided in the methodology section for better understanding and article flow.”
    Response: We acknowledge the reviewer’s suggestion. While we did not introduce separate subheadings, we revised and clarified the text of the Materials and Methods section to improve flow and readability. Key methodological aspects—such as patient selection, inclusion criteria for IOUS, surgical technique, outcome variables, and statistical analysis—are now described in a clearer and more structured narrative format.
  2. “In methodology, add the ethical approval number obtained.”
    Response: This was included:
    “The study was approved by the institutional review board (approval code: 06.05.2025).”

  1. “The result section does not clearly explain the main findings.”
    Response: The Results section was revised for clarity. We emphasized that IOUS was used significantly more often in complex cases (75% in Group A vs. 10% in Group B), yet perioperative outcomes such as warm ischemia time and complication rates remained comparable, and PSM rates were equally low across groups.

  1. “Statistical analysis is based on very few cases. This major limitation should be discussed.”
    Response: This limitation was explicitly added in the Discussion:
    “Another major limitation is that statistical power was limited by the small number of cases, especially for oncological outcomes such as positive surgical margins. This restricts the ability to demonstrate a statistically significant independent effect of IOUS.”

  1. “The discussion looks like a review of IOUS, but it does not describe the interpretation of the study results. The discussion is iterating the points already stated in the introduction.”
    Response: We revised the Discussion to highlight the study-specific findings:
    “Our results show that IOUS was preferentially used in patients with higher surgical complexity, yet perioperative and functional outcomes were not inferior compared to non-IOUS cases. This suggests that IOUS may compensate for the technical challenges of complex tumors, supporting its role as a practical intraoperative adjunct.”

  1. “The limitations and novelty of the study should be explained in detail in the discussion.”
    Response: We expanded the Limitations section to emphasize both constraints and novelty:
    “The retrospective design, small sample size, and selective use of IOUS limit the generalizability of our findings. However, the novelty of this study lies in evaluating IOUS across both robotic and open partial nephrectomies, including solitary kidneys and recurrent tumors, providing insight into scenarios where IOUS may be most beneficial.”

  1. “In conclusion, IOUS is declared as a standard of care. Rather, it would be better to state IOUS as a promising approach by stating its future prospects.”
    Response: The Conclusion was revised accordingly:
    “IOUS represents a promising adjunct in complex partial nephrectomy, enhancing tumor localization and supporting parenchymal preservation. Future multicenter prospective studies are required to validate its role before routine adoption as standard of care.”

Round 2

Reviewer 2 Report

Comments and Suggestions for Authors

The authors have revised the manuscript, appropriately reframing it as an exploratory analysis. They now acknowledge that, although IOUS appears promising, the present study does not provide sufficient evidence to demonstrate its efficacy. The discussion and limitations have been suitably revised.

I would, however, recommend adding units in Table 2 for blood loss, tumor size, and length of stay.

Author Response

We sincerely thank the reviewer for the careful evaluation of our revised manuscript and the constructive feedback provided.

Comment: I would, however, recommend adding units in Table 2 for blood loss, tumor size, and length of stay.

Response: We fully agree with the reviewer’s suggestion. Units have now been added to Table 2 as follows:

  • Blood loss: milliliters (mL)

  • Tumor size: centimeters (cm)

  • Length of stay: days (d)

The revised version of Table 2 has been updated accordingly in the manuscript.

We appreciate the reviewer’s positive assessment regarding the reframing of the manuscript, the improved discussion, and acknowledgment of study limitations. We are confident that these refinements further enhance the clarity and rigor of our work.

Reviewer 3 Report

Comments and Suggestions for Authors

May be accepted for publication.

Author Response

We sincerely thank the reviewer for the constructive feedback and for considering our manuscript for publication.

Comment: May be accepted for publication.

Response: We greatly appreciate the reviewer’s positive evaluation and support for the acceptance of our work.